# Protective effects of dapagliflozin against oxidative stress-induced cell injury in human proximal tubular cells

Nawel Zaibi[1], Pengyun Li[1¤], Shang-Zhong Xu[1,2]*

1 Centre for Atherothrombosis and Metabolic Disease, Hull York Medical School, University of Hull, Hull, United Kingdom, 2 Diabetes, Endocrinology and Metabolism, Hull York Medical School, University of Hull, Hull, United Kingdom

¤ Current address: Key Lab of Medical Electrophysiology, Ministry of Education, Institute of Cardiovascular Research, Southwest Medical University, Luzhou, China
* sam.xu@hyms.ac.uk

**Data Availability Statement:** All relevant data are within the manuscript and its Supporting Information files. We have given the procedures, part of raw data, analysis and graphs on the protocol hub: https://www.protocols.io/view/

## Abstract

Elevated reactive oxygen species (ROS) in type 2 diabetes cause cellular damage in many organs. Recently, the new class of glucose-lowering agents, SGLT-2 inhibitors, have been shown to reduce the risk of developing diabetic complications; however, the mechanisms of such beneficial effect are largely unknown. Here we aimed to investigate the effects of dapagliflozin on cell proliferation and cell death under oxidative stress conditions and explore its underlying mechanisms. Human proximal tubular cells (HK-2) were used. Cell growth and death were monitored by cell counting, water-soluble tetrazolium-1 (WST-1) and lactate dehydrogenase (LDH) assays, and flow cytometry. The cytosolic and mitochondrial (ROS) production was measured using fluorescent probes (H2DCFDA and MitoSOX) under normal and oxidative stress conditions mimicked by addition of $H_2O_2$. Intracellular $Ca^{2+}$ dynamics was monitored by FlexStation 3 using cell-permeable $Ca^{2+}$ dye Fura-PE3/AM. Dapagliflozin (0.1–10 µM) had no effect on HK-2 cell proliferation under normal conditions, but an inhibitory effect was seen at an extreme high concentration (100 µM). However, dapagliflozin at 0.1 to 5 µM showed remarkable protective effects against $H_2O_2$-induced cell injury via increasing the viable cell number at phase G0/G1. The elevated cytosolic and mitochondrial ROS under oxidative stress was significantly decreased by dapagliflozin. Dapagliflozin increased the basal intracellular $[Ca^{2+}]_i$ in proximal tubular cells, but did not affect calcium release from endoplasmic reticulum and store-operated $Ca^{2+}$ entry. The $H_2O_2$-sensitive TRPM2 channel seemed to be involved in the $Ca^{2+}$ dynamics regulated by dapagliflozin. However, dapagliflozin had no direct effects on ORAI1, ORAI3, TRPC4 and TRPC5 channels. Our results suggest that dapagliflozin shows anti-oxidative properties by reducing cytosolic and mitochondrial ROS production and altering $Ca^{2+}$ dynamics, and thus exerts its protective effects against cell damage under oxidative stress environment.

protocol-for-intracellular-ca2-detection-using-fle-bnubmesn.

**Funding:** This project has received funding from the Innovative Medicines Initiative 2 Joint Undertaking under grant agreement No 115974. Funder has no role in the study design and data collection and manuscript preparation.

**Competing interests:** No authors have competing interests.

# Introduction

Dapagliflozin, also named Farxiga®, is a glucose-lowering therapeutic drug in the gliflozins family [1,2]. It was developed by Bristol-Myers in partnership with AstraZeneca and approved by FDA in 2014 [3]. Dapagliflozin blocks sodium glucose co-transporter 2 (SGLT-2) encoded by *SLC5A2* gene and thus inhibits the reabsorption of filtered glucose and sodium [4,5]. Recently, clinical trials have demonstrated that the inhibition of glucose uptake by proximal tubules not only leads to lower glucose and glycated haemoglobin (HbA1c) levels in the blood but also reduces the risk of developing later life diabetic complications, such as cardiovascular events and diabetic kidney disease [2,6,7]. However, such beneficial effects and underlying mechanisms of SGLT-2 inhibitors are largely unknown.

Oxidative stress is a main factor for causing cell injury in diabetes. There are very few reports on the effect of dapagliflozin under oxidative stress conditions. In db/db mice, treatment with dapagliflozin has been shown to reduce renal expression of some enzymes related to reactive oxygen species (ROS) production and inflammation including NF-kB, Nox2, Nox4, TGFβ1, monocyte chemoattractant protein-1 (MCP-1), osteopontin, intercellular adhesion molecule-1 (ICAM-1), and myeloperoxidases (MPO) [8–10]. The reduction of ROS production was also observed in rat heart after a 10-week treatment with the dapagliflozin analogue empagliflozin [11]. These studies suggest that SGLT-2 inhibitors may have anti-oxidative properties. In addition, dapagliflozin decreases the amplitude of intracellular $Ca^{2+}$ transient and L-type $Ca^{2+}$ current in cardiac myocytes isolated from normal and diabetic rats [12], suggesting it may affect intracellular calcium dynamics.

Here we have investigated the effects of dapagliflozin on cell growth and death under normal and oxidative stress environment that mimics diabetic conditions. The effects on cytosolic and mitochondrial ROS production and intracellular $[Ca^{2+}]_i$, endoplasmic reticulum (ER) $Ca^{2+}$ release, and store-operated $Ca^{2+}$ influx have also been examined in the human proximal tubular cells (HK-2), since this cell type has specific expression of the drug targeted protein SGLT-2. The anti-oxidative stress effect we reported here could be an alternative mechanism for the explanation of the beneficial effects of SGLT-2 inhibitors.

# Materials and methods

## Cell culture and transfection

The HK-2 cell line was purchased from LGC standards (Catalogue number CRL-2190, UK). HK-2 cells were maintained in DMEM/F-12 medium with 5 mM glucose and supplemented with 10% foetal calf serum (FCS), 10 mM HEPES and 100 units·mL$^{-1}$ penicillin and 100 μg·mL$^{-1}$ streptomycin. The function of reabsorption for the HK-2 cell line was characterized in our previous report [13]. The inducible TRPM2 cells were generated by transfection of human TRPM2 gene (GenBank accession number BC112342) in pcDNA4/TO tetracycline-regulatory vector into HEK-293 T-REx cells (Invitrogen, Paisley, UK). The TRPM2 cells were cultured and maintained using standard DMEM/F-12 medium. The expression of TRPM2 was induced by tetracycline (1 μg·mL$^{-1}$) and the function was characterized as we described previously [14]. All the cell cultures were maintained at 37˚C under 95% air and 5% $CO_2$ without mycoplasma contamination.

## Cell apoptosis, necrosis and proliferation assays

Apoptosis was measured by direct cell counting after Hoechst 33342 and propidium iodide nuclear staining. HK-2 cells were fixed with 4% paraformaldehyde in 100 mM phosphate buffer solution (22.6 mM $NaH_2PO_4$ and 77.4 mM $Na_2HPO_4$), washed out with PBS, and

incubated with Hoechst 33342 (1 μM) and propidium iodide (15 μM) for 30 min in the dark. The cells were washed with PBS twice and the nuclear staining was photographed using a fluorescent microscopy. The apoptotic cells with condensed nuclear staining were counted using CellC software. For necrotic cell death, the activity of lactate dehydrogenase (LDH) in the culture medium that released from the cytosol was determined using a Cytotoxicity Detection Kit (Roche) with similar procedures in our previous report [15]. Cell proliferation was determined using a water-soluble tetrazolium-1 (WST-1) assay in which tetrazolium salts are cleaved by mitochondrial dehydrogenase to form formazan in viable cells [16]. The absorbance for WST-1 and LDH assays was measured using a spectrophotometer.

## Fluorescence-activated cell sorting (FACS)

The HK-2 cells were seeded into a 6-cm petri dish at a confluence of 5000 cells/mL and incubated in a humidified atmosphere of 5% $CO_2$ and 95% air at 37˚C for 24 hours. The cells were pre-treated with different concentrations of dapagliflozin for 2 hours before addition of 200 μM $H_2O_2$, followed by a 24-hour incubation with dapagliflozin or/and $H_2O_2$. The cells were trypsinised with 0.25% trypsin-EDTA and centrifuged twice with PBS in FACS tubes at 300 g for 5 minutes. The PBS was then removed and 200 μL of 10 μg/mL propidium iodide was added to all the tubes and incubated for 15 minutes before mounting for FACS detection. The cell cycle was analysed using CellQuest software and all groups were set in triplicates.

## Cytosolic and mitochondrial ROS assays

The cytosolic ROS fluorescent indicator H2DCFDA was used to monitor ROS production. Briefly, HK-2 cells were seeded into a 96-well plate at a confluence of $6 \times 10^4$ cells per well and incubated for 24 hours. The cells were washed twice with warm PBS and loaded with H2DCFDA dye at a final concentration of 2 μM in standard bath solution. The standard bath solution contained (mM): 130 NaCl, 5 KCl, 8 D-glucose, 10 HEPES, 1.2 $MgCl_2$, and 1.5 $CaCl_2$, and the pH was adjusted to 7.4 with NaOH. After incubation for 45 minutes at 37˚C in the dye, cells were washed with standard bath solution and dapagliflozin at different concentrations was added into the wells to a final volume of 200 μL. The plate was then placed in FlexStation 3 (Molecular Device, USA) and the fluorescence intensity was read at excitation/emission wavelength of 492/520 nm using a kinetics protocol. The protocol was set to automatically add 10 μL of 2 mM $H_2O_2$ solution at a time point of 300 seconds to give a final concentration of 200 μM $H_2O_2$.

For mitochondrial ROS assay, the mitochondrial ROS indicator, MitoSOX, was loaded at a concentration of 5 μM in PBS. MitoSOX red is a cationic derivative of dihydroethidium which enters live cells and specifically intercalates with mitochondrial DNA and results in red fluorescence. The successful loading of MitoSOX was examined using a fluorescent microscopy and the cells were then washed twice with standard bath solution. Dapagliflozin at different concentrations was added to the cells for 3–5 min prior to the experiment. The plate was then placed in FlexStation 3 and fluorescence was measured at excitation/emission wavelength of 510/580 nm.

## $[Ca^{2+}]_i$ measurement

Cells were pre-incubated with 2 μM Fura-PE3/AM at 37˚C for 30 min in $Ca^{2+}$-free bath solution (mM):130 NaCl, 5 KCl, 1.2 $MgCl_2$, 10 HEPES, 8 D-glucose, and 0.4 EGTA, followed by a 20-min wash period in the standard bath solution at room temperature. The ratio of $Ca^{2+}$ dye fluorescence ($F_{340}/F_{380}$) was measured using FlexStation 3. Thapsigargin (TG, 1μM) was used to block SERCA in the ER to cause ER $Ca^{2+}$ release signal in $Ca^{2+}$-free bath solution. The

store-operated $Ca^{2+}$ entry (SOCE) was measured by addition of 1.5 mM $Ca^{2+}$ after the ER $Ca^{2+}$ store depletion by TG.

## Electrophysiology

The whole-cell patch clamp procedure is similar to our previous report [17]. Briefly, electrical signal was amplified with an Axopatch 200B patch clamp amplifier and controlled with pClamp software 10. A 1-s ramp voltage protocol from –100 mV to +100 mV was applied at a frequency of 0.2 Hz from a holding potential of 0 mV. Signals were sampled at 1 kHz. The glass microelectrode with a resistance of 3–5 MΩ was used. The pipette solution contained (in mM): 115 CsCl, 10 EGTA, 2 $MgCl_2$, 10 HEPES and 5.7 $CaCl_2$, pH was adjusted to 7.2 with CsOH and osmolarity was adjusted to ∼290 mOsm with mannitol, and the calculated free $Ca^{2+}$ was 200 nM. ADP-ribose (0.5 mM) was added in the pipette solution to activate TRPM2. The cells were perfused with standard bath solution during whole-cell patch recording. The experiment was performed at room temperature (25˚C).

## RT-PCR

Total RNA was extracted from the HK-2 cells using TRI reagent (Sigma). The RNA was reverse transcribed with moloney murine leukaemia virus reverse transcriptase (RT) using random primers and oligo dT primers (Promega). Quantitative RT-PCR was performed using StepOne™ Real-Time PCR System (Applied Biosystems, UK). The primer set was designed across introns to avoid genomic DNA contamination and synthesized by Sigma-Genosys. The primer sequences were: SGLT-1 (ACCTTTCCCTTCTGTCCCTG and CATGATCACCGTCTGCAAGG); SGLT-2-primer set 1 (GACATGTTCTCCGGAGCTGT and GCTCCCAGGTATTTGTCGAA); SGLT-2-primer set 2 (TCACGATGCCACAGTACC and TAGATGTTCCAGCCCAGAGC); β-actin (ACAGAGCCTCGCCTTTGC and GGAATCCTTCTGACCCATGC). Each reaction volume was 10 μl, which contained 1 × SYBR Green PCR master mix (Applied Biosystems), 5 μl of cDNA, 0.75 μl of 300 nM forward primer, and 0.75 μl of 300 nM reverse primer. Human β-actin was used as an internal standard for control. The PCR cycle was programmed as an initial cycle of 50 ˚C for 2 min, followed by 95 ˚C for 10 min, then 50 repeated cycles of 95 ˚C for 15 s denaturation and 54 ˚C annealing temperature for 30 s, and primer extension at 72 ˚C for 30 s. The PCR products were examined by electrophoresis on 2% agarose gels.

## Chemicals and reagents

All general salts and reagents were from Sigma (UK). Dapagliflozin, hydrogen peroxide ($H_2O_2$), 2-aminoethoxydiphenyl borate (2-APB), tetracycline, thapsigargin, propidium iodide, Hoechst 33342 and FCCP (carbonyl cyanide 4-(trifluoromethoxy)phenylhydrazone) were purchased from Sigma-Aldrich. MitoSOX and H2DCFDA were purchased from Thermo Fisher Scientific UK. Fura-PE3/AM was purchased from Invitrogen (UK). Dapagliflozin (100 mM), Fura-PE3/AM (1 mM), thapsigargin (1 mM) and 2-APB (100 mM) were made up as stock solutions in 100% dimethyl sulphoxide (DMSO).

## Statistics

Data are expressed as mean ± SEM. Where *n* is the number of wells for FlexStation experiments or cell number for electrophysiological recordings. Mean data were compared using paired *t* test for the results before and after treatment, or unpaired *t* test for two group comparison. ANOVA with Bonferroni's post-hoc analysis was used for comparing more than two groups with significance indicated if $P < 0.05$.

## Results

### Expression of SGLT isoforms in HK-2 cells

HK-2 cell line is an immortalized proximal tubule epithelial cell line derived from normal adult human kidney [13,18]. To confirm the phenotype of human proximal tubular cell line, the expression of sodium glucose co-transporter-1 (SGLT-1) and -2 (SGLT-2) was examined using RT-PCR. Two primer sets across introns were used for SGLT-2 and β-actin was set as a positive control, the bands for SGLT-2 were much stronger than that for SGLT-1 and β-actin (Fig 1), suggesting that SGLT-2 is highly expressed in the proximal tubular cell line, which also confirms the specific cell marker for the proximal tubular cells used in the study.

### Oxidative stress-induced cell injury

Cell growth and death were examined under oxidative stress conditions mimicked by addition of $H_2O_2$. Cell proliferation was monitored by WST-1 assay, which is proportional to the number of viable cells. HK-2 cells were incubated with different concentrations of $H_2O_2$ for 24 hours and the cell proliferation was significantly decreased in a concentration-dependent manner (Fig 2A). The half-maximal inhibitory concentration ($IC_{50}$) was calculated according to the normalised dose response curve with an $IC_{50}$ of 115 μM and a slope factor of 1.74 (Fig 2B).

The effect of $H_2O_2$ on HK-2 cell death was determined by LDH assay after incubation with different concentrations of $H_2O_2$ ranging from 0 to 4000 μM for 24 hours (Fig 2C). The LDH activity in the medium increased in a concentration-dependent manner. The $IC_{50}$ was 199 μM with a slope factor (p) of 0.007 (Fig 2D), suggesting that $H_2O_2$ significantly induced cell death.

To further investigate $H_2O_2$-induced cell death, we examined the apoptotic cell death using nuclear staining with propidium iodide and Hoechst 33342. After 24 hours incubation with 200 μM $H_2O_2$, the apoptotic cell number was counted after taking pictures using a fluorescent microscopy. The HK-2 cells treated with $H_2O_2$ showed a significant increase of apoptotic cell death (Fig 2C and 2D). These results suggest that $H_2O_2$ is a useful agent to induce human proximal tubular cell injury to mimic oxidative stress condition.

### Effects of dapagliflozin on cell proliferation and death under normal and oxidative stress conditions

To investigate the effects of dapagliflozin on HK-2 cell proliferation under normal and oxidative stress conditions, cells were treated with different concentrations of dapagliflozin for 24-hour with or without incubation with 200 μM $H_2O_2$. $H_2O_2$ at 200 μM was used because this concentration near the $IC_{50}$ can mimic oxidative stress to cause cell growth inhibition and cell death. Under normal cell culture condition, dapagliflozin itself did not affect cell proliferation except the highest concentration (100 μM) that showed a significant decrease in cell proliferation, suggesting a potential cytotoxic effect at the very high concentration. However, dapagliflozin at concentrations between 0.1 and 10 μM showed a significant increase in proliferation under the oxidative stress condition by adding 200 μM $H_2O_2$. Dapagliflozin at 0.1 μM showed the maximum protective effect against the $H_2O_2$-induced inhibition of cell proliferation.

The effect of dapagliflozin on cell death was also examined. In normal condition, dapagliflozin at 0.1, 1 and 5 μM did not show any significant effects on LDH release from the HK-2 cells, but higher concentrations (10 and 100 μM) showed a significant increase compared to the control (Fig 3B), suggesting that the drug at high concentrations can cause cell injury, which should be alerted in clinic practice. In the cells treated with $H_2O_2$, dapagliflozin at 0.1, 1, and 5 μM showed a significant reduction in LDH release compared to the control. The greatest protection occurred at the concentration of 0.1 μM (Fig 3B). These results suggested that

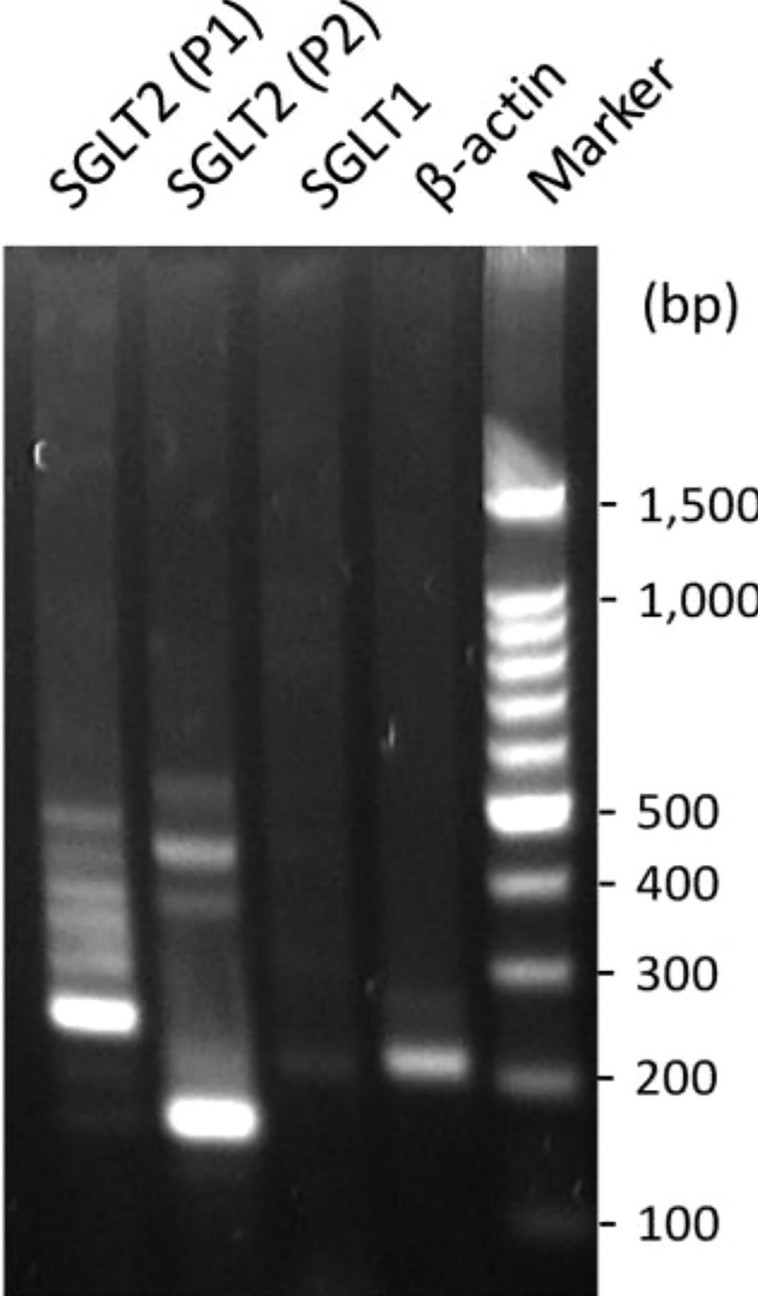

**Fig 1. The mRNAs of SGLT-1 and SGLT-2 were detected in HK-2 cells by RT-PCR.** The β-actin was set as positive control. Two primer sets (P1 and P2) were used for SGLT-2. The product size for β-actin, SGLT-1, SGLT-2 (P1) and SGLT-2 (P2) is 203 bp, 211 bp, 236 bp, and 159 bp, respectively. 2% agarose gel was used (Also see full length gel in S1 Fig).

dapagliflozin is a potent agent in the protection of oxidative stress-induced cell death in proximal tubular cells. Similar to cell proliferation, too high concentration of dapagliflozin (100 μM) did not show any protection but damage the cells.

The apoptotic cells were counted using nuclear staining methods with propidium iodide and Hoechst 33342. Dapagliflozin at 0.1 μM significantly reduced the percentage of apoptotic

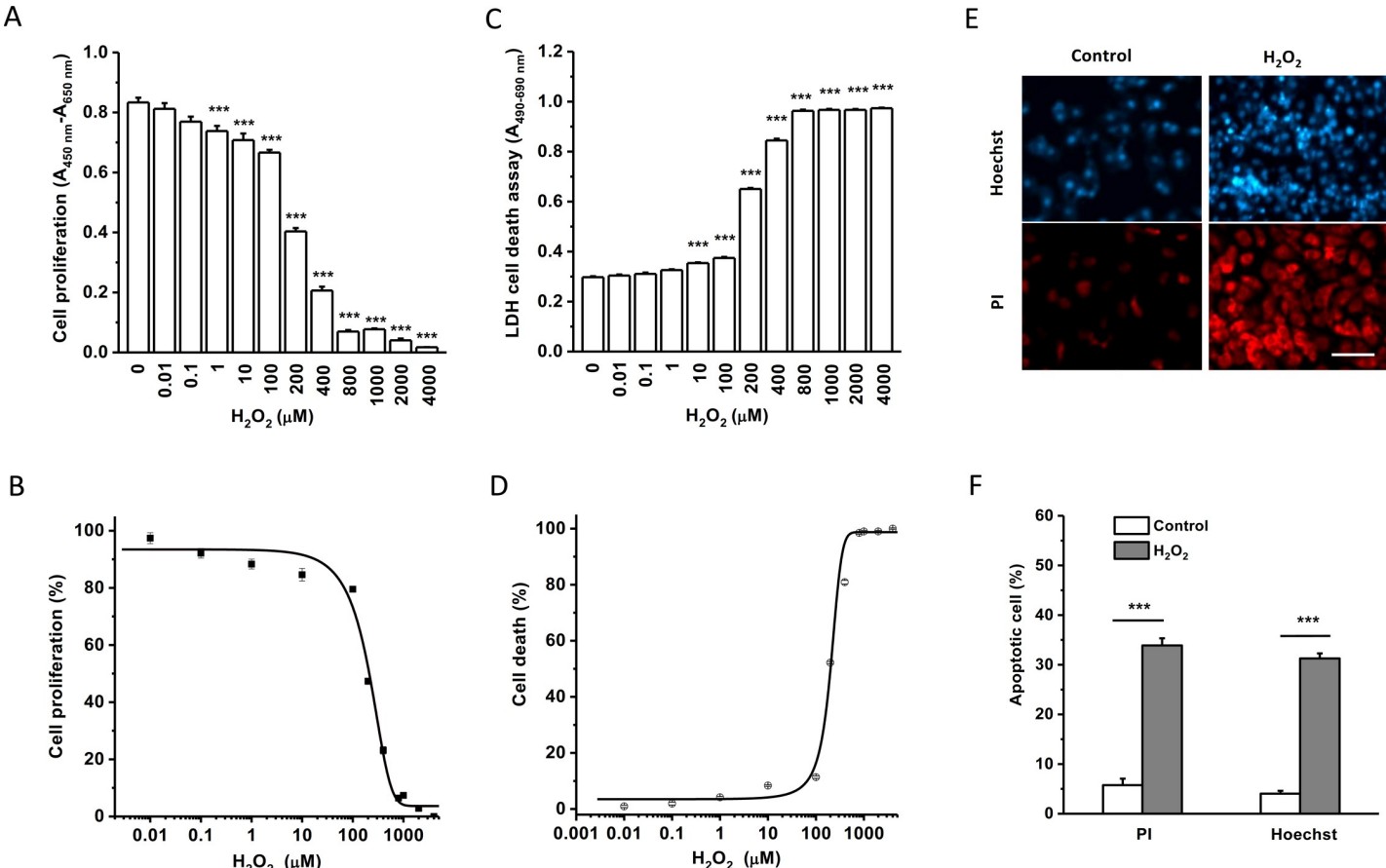

**Fig 2. Cell proliferation and death under H$_2$O$_2$-induced oxidative stress conditions.** The proliferation of HK-2 cells was assayed with WST-1 reagents after a 24-hour incubation with different concentrations of H$_2$O$_2$. The absorbance was measured at a wavelength of 450 nm with a reference wavelength of 650 nm. A, Mean ± SEM data ($n$ = 8 for each group). B, The normalised dose-response data was fitted with the equation: y = A2 + (A1-A2)/(1 + (x/x0)^p), and the IC$_{50}$ was 114.9 μM. C, The cytotoxicity of HK-2 cells was assayed using LDH reagents after a 24-hour incubation with H$_2$O$_2$, and the absorbance was measured at a wavelength of 490 nm and a reference wavelength of 680 nm. D, The dose–response curve showed an IC$_{50}$ of 199.4 μM for cytotoxicity ($n$ = 8 for each group, ANOVA with Bonferroni post hoc test was used for group comparison). E, Examples of Hoechst 33342 (Hoechst) and propidium iodide (PI) nuclear staining. Scale bar = 100 μm. F, Mean ± SEM data for apoptotic cell counting ($n$ = 6 for each group). $^{***}$ $P<0.001$.

cell death (Fig 3C and 3D), which further supports the protective effects of dapagliflozin against H$_2$O$_2$-induced cell death.

## Cell cycle response to H$_2$O$_2$ and the effect of dapagliflozin

The effects of dapagliflozin on H$_2$O$_2$-induced cell cycle arrest have been examined. The HK-2 cells treated with different concentrations of dapagliflozin with or without H$_2$O$_2$ and the effect on cell cycle was analysed by FACS after staining with propidium iodide (Fig 4A). Cells treated with 200 μM H$_2$O$_2$ showed a significant increase of cells arrested in the G2/M phase (~ 33%) and in the S phase (6%) compared to the non-treated control, suggesting that the cell cycle of HK-2 cells is arrested by the oxidative stress condition. In the group without H$_2$O$_2$ treatment, dapagliflozin did not significantly alter the cell cycle except the high concentration group (10 μM) with an increased percentage of cells in S phase, reflecting an active repair response (Fig 4B). In cells treated with both dapagliflozin and 200 μM H$_2$O$_2$, dapagliflozin at 0.01 and 0.1 μM significantly increased the percentage of cells in G0/G1 phase and decreased the percentage of cells in G2/M phase. Dapagliflozin at 0.1 μM also reduced the percentage of cells in

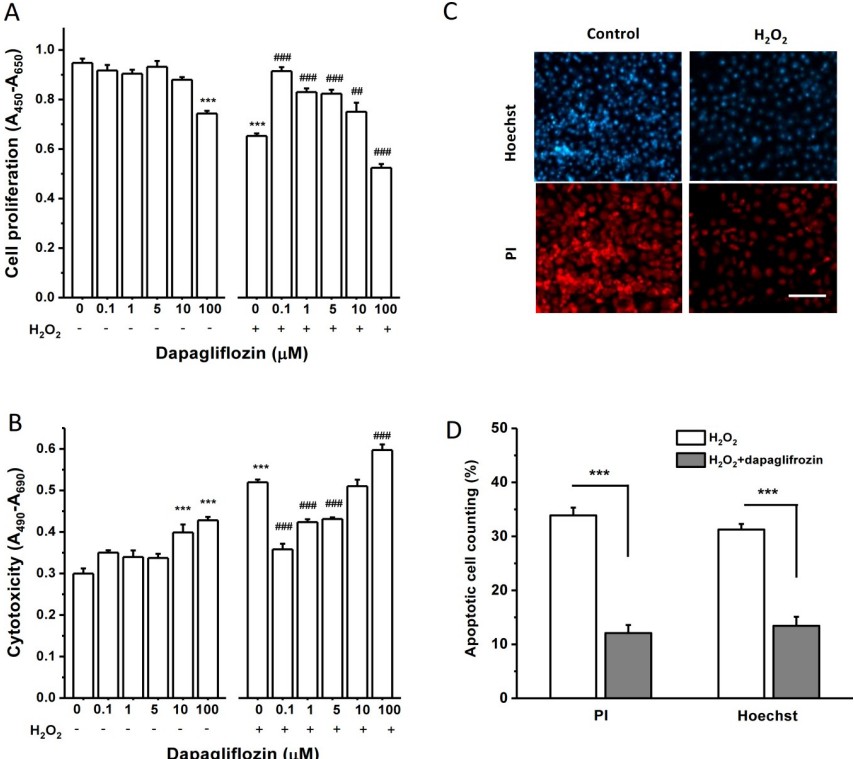

**Fig 3. Effects of dapagliflozin on cell growth and death under normal and H$_2$O$_2$-induced oxidative stress conditions.** A, Effect on cell proliferation under normal and oxidative stress conditions mimicked with 200 μM H$_2$O$_2$. Cell proliferation was assayed with WST-1 reagents after incubation with dapagliflozin for 24 hours. The absorbance was measured ($n$ = 8 for each group). B, Cytotoxicity was assayed with LDH reagents after incubation with dapagliflozin for 24 hours, with or without 200 μM H$_2$O$_2$ ($n$ = 8 for each group). The absorbance was measured at a wavelength of 490 nm and a reference wavelength of 680 nm. In A and B, *** $P<0.001$ is for the comparison with control, and ## $P<0.01$ and ### $P<0.001$ for the comparison with H$_2$O$_2$ control group without dapagliflozin. One-way ANOVA with Bonferroni post hoc test was used for group comparisons. C, Examples of nuclear staining with propidium iodide (PI) and Hoechst 33342 (Hoechst). Scale bar = 100 μm. D, Mean ± SEM data for the percentage of apoptotic cells in the group of H$_2$O$_2$ (200 μM) and H$_2$O$_2$ (200 μM) plus dapagliflozin (0.1 μM). ($n$ = 6 for each group, *** $P<0.001$ by unpaired $t$-test).

the S phase (Fig 4B). These results suggest that the alteration of cell cycle by oxidative stress can be prevented by dapagliflozin.

## Dapagliflozin prevents H$_2$O$_2$-induced cytosolic ROS production

The cytosolic ROS production was real-time monitored using FlexStation 3 in normal and oxidative stress conditions using fluorescent ROS indicator H2DCFDA. After adding 200 μM H$_2$O$_2$ to the cells, the fluorescence intensity gradually increased and the slope was much bigger than the control group without H$_2$O$_2$ (Fig 5A and 5B), suggesting that the ROS production is higher in the oxidative stress condition. This result is consistent with the previous report [19], suggesting that H$_2$O$_2$ stimulates cytosolic ROS production.

To investigate the effects of dapagliflozin on ROS production, HK-2 cells were incubated with different concentrations of dapagliflozin. The fluorescence was gradually increased as the incubation time increasing, however there was no statistical difference compared to the control (Fig 5C and 5D), suggesting that dapagliflozin (0.1–10 μM) did not affect cytosolic ROS production in normal condition. However, under oxidative stress conditions, dapagliflozin at 0.1 μM showed a significant reduction of ROS production compared to the H$_2$O$_2$ control,

reflecting its antioxidative properties. Nevertheless, dapagliflozin at 1 and 10 μM did not reduce ROS (Fig 5E and 5F), suggesting the beneficial effect of dapagliflozin is associated with the concentrations used.

## Mitochondrial ROS production reduced by dapagliflozin

To observe the effects on mitochondrial ROS, MitoSOX was used as mitochondrial superoxide indicator. Cells were incubated with MitoSOX (5 μM) for 30 min and the red fluorescence in mitochondria was evident (Fig 6A). The intensity of MitoSOX was significantly increased after incubation with a mitochondrial oxidative phosphorylation uncoupler FCCP (Fig 6B), suggesting that MitoSOX is specific for mitochondrial ROS detection. $H_2O_2$ at 200 μM also significantly increased mitochondrial ROS production (Fig 6C and 6D). These data suggest that mitochondrial ROS production was enhanced in the cells under oxidative stress conditions.

The effects of dapagliflozin on mitochondrial ROS were examined in normal and oxidative stress conditions. MitoSOX fluorescence intensity was not changed significantly compared to the control, suggesting that dapagliflozin did not affect mitochondrial ROS production in normal condition. Under oxidative stress condition dapagliflozin at 0.1 μM significantly decreased the ROS production in the mitochondria (Fig 6G and 6H), suggesting its preventive effect on mitochondrial ROS production. However, higher concentrations (1, 10 μM) of dapagliflozin did not show any protection.

## Intracellular calcium dynamics modulated by dapagliflozin

The interplay of ROS and intracellular $[Ca^{2+}]_i$ is an important cellular process for regulating mitochondrial function, gene expression and apoptosis [20], therefore the $[Ca^{2+}]_i$ was

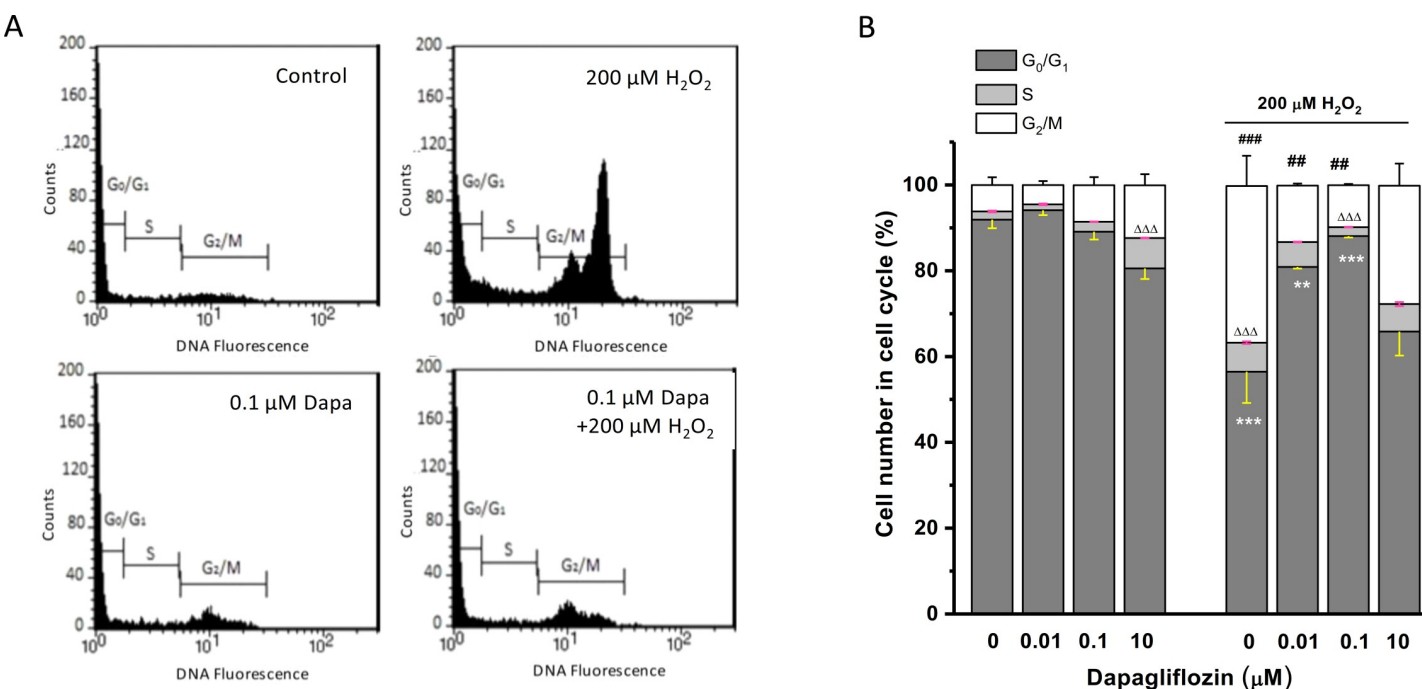

**Fig 4. Cell cycle response to oxidative stress and the effect of dapagliflozin.** A, Human proximal tubular cells (HK-2) were cultured in 6-cm petri dishes and the oxidative stress was induced with 200 μM $H_2O_2$ for 24 hours. The histogram examples of the cells stained with propidium iodide with or without 0.1 μM dapagliflozin (Dapa). B, Cell cycle was quantified by FACS and mean ± SEM data were presented ($n$ = 3). One-way ANOVA with Bonferroni post hoc test was used for group comparison. ** $P$<0.01 and *** $P$<0.001 for G0/G1 phase comparison; ΔΔΔ $P$<0.001 for S phase comparison; and ## $P$<0.01 and ### $P$<0.001 for G2/M phase comparison.

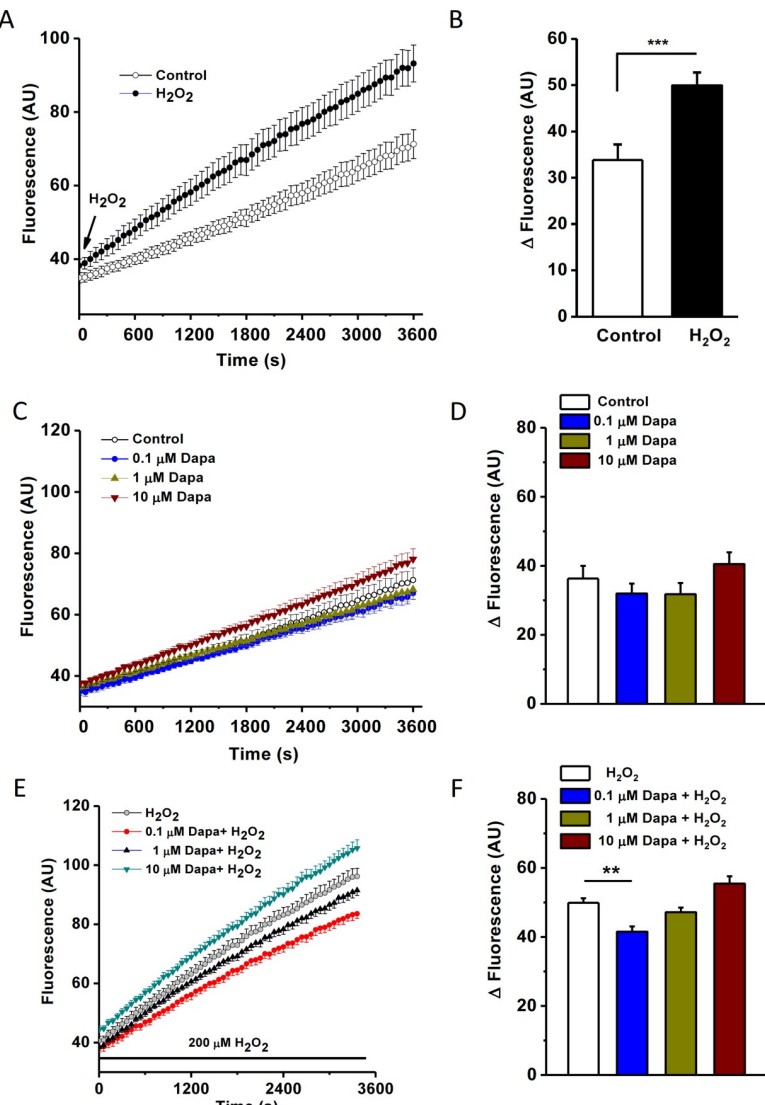

**Fig 5. Cytosolic ROS production under oxidative stress and the protection by dapagliflozin.** HK-2 cells were loaded with cytosolic ROS indicator, H2DCFDA (2 μM) and the fluorescence intensity was dynamically monitored using FlexStation at an excitation wavelength of 492 nm and an emission wavelength of 520 nm. Reads were sampled every 60 seconds. A. The time course of fluorescence intensity in control and $H_2O_2$-treated groups. B, Change of cytosolic ROS fluorescence in 60 min for the control (without $H_2O_2$) and $H_2O_2$ groups ($n = 5$, *** $P < 0.001$). C, Effects of dapagliflozin (Dapa) on cytosolic ROS production under normal conditions. D, Mean ± SEM data for the change of fluorescence in HK-2 cells within 60 min. E, Time course of cytosolic ROS indicator fluorescence for the cells treated with $H_2O_2$ (200 μM) and different concentrations of dapagliflozin. F, Mean ± SEM data for the delta fluorescence changes in 60 min ($n = 5$ for each concentration). One-way ANOVA with Bonferroni post hoc test was used. ** $P < 0.01$.

measured by FlexStation 3 using the $Ca^{2+}$ dye Fura-PE3/AM. The HK-2 cells were treated with different concentrations of dapagliflozin and the basal $Ca^{2+}$ fluorescence was determined under normal bath solution. Dapagliflozin at concentrations ranging from 0.01 μM to 10 μM increased basal intracellular calcium level (Fig 7A). To explore the underlying mechanisms, the ER calcium release and SOCE were examined, and dapagliflozin had no significant effects on TG-induced $Ca^{2+}$ release and SOCE (Fig 7B and 7C). We also examined the effect on $H_2O_2$-sensitive cation channel TRPM2. Interestingly, pre-incubation with dapagliflozin

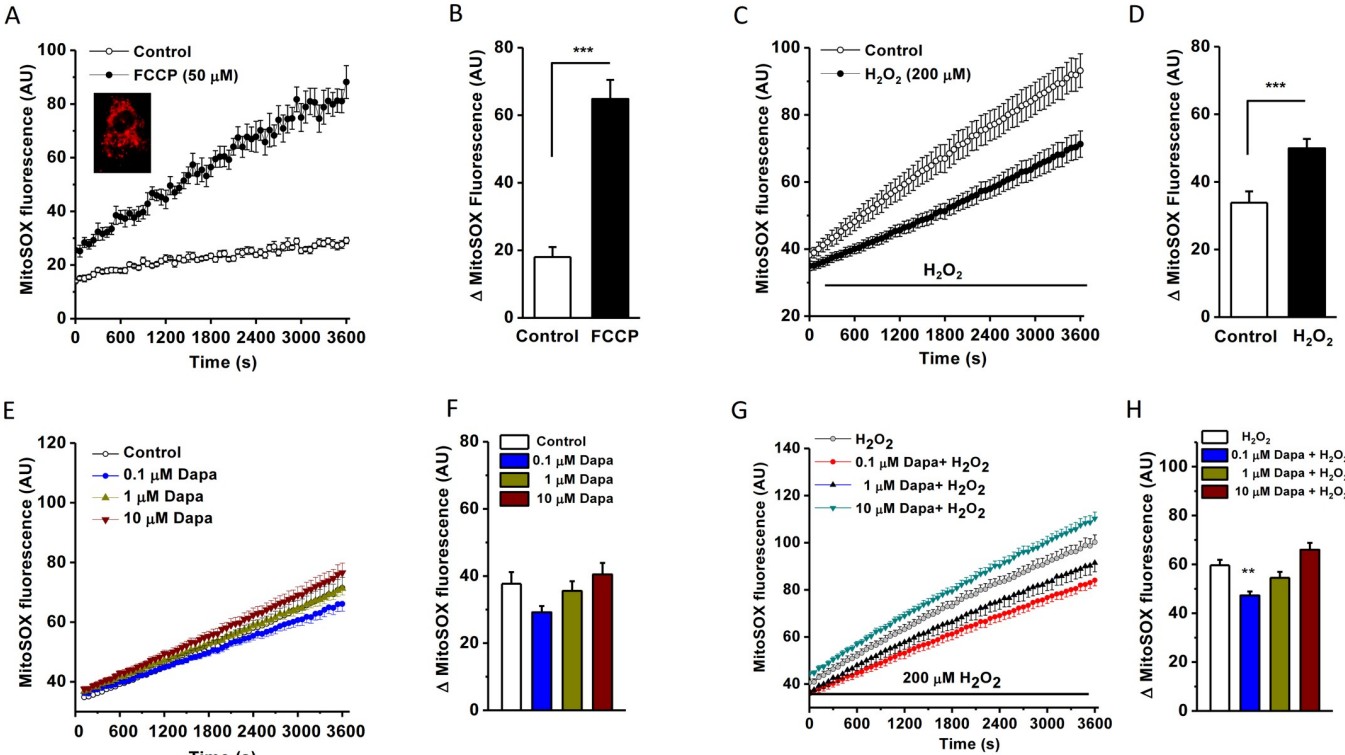

**Fig 6. Mitochondrial ROS detected by MitoSOX and the effect of dapagliflozin.** A. HK-2 cells loaded with mitochondrial ROS indicator MitoSOX (5 μM) and measured by FlexStation 3 at the excitation wavelength of 510 nm and emission wavelength of 580 nm and the kinetics was sampled every 60 sec. The time course for the cells treated with FCCP (50 μM) or without FCCP (control). Example of MitoSOX (Red) loading was shown in the inset. B. Mean data for MitoSOX intensity change at the time point of 60 min in both control and FCCP groups ($n = 8$ for each group, *** $P<0.001$). C, Time course for MitoSOX fluorescence in the cells treated with or without (control) 200 μM $H_2O_2$. D, Mean ± SEM data ($n = 7$ for each group, *** $P<0.001$ assessed by unpaired $t$-test). E, Effect of dapagliflozin (Dapa) on the intensity of mitochondrial ROS fluorescence in HK-2 cells under normal conditions. F, Mean ± SEM data ($n = 7$ for each group). G, Effect of dapagliflozin under $H_2O_2$-induced oxidative stress conditions. H, Mean ± SEM data for the delta changes of mitochondrial MitoSOX intensity within 60 min treatment ($n = 7$ for each group). One-way ANOVA with Bonferroni *post hoc* test was used, ** $P<0.01$.

significantly enhanced the $Ca^{2+}$ influx in the Tet-induced TRPM2 cells with a unique second phase of activation, but no such phenomena in the non-induced control cells (Fig 7D). This delayed activation could be due to an indirect effect, since dapagliflozin had no direct effect on the channel per se using whole-cell patch clamp recording with intracellular ADP-ribose activator (Fig 7F). Dapagliflozin had no effects on TRPC4, TRPC5, ORAI1 and ORAI3 (Fig 8), and those channels are also expressed in the human proximal tubular cells [13,21–24] and have redox-sensitivity [25–27].

## Discussion

Our results show that dapagliflozin displays a significant protective effect against oxidative stress-induced cell injury in HK-2 cells. The elevated cytosolic ROS and mitochondrial ROS production under oxidative stress conditions is reduced by pre-treatment with dapagliflozin. Dapagliflozin also increases basal intracellular $Ca^{2+}$ level and interferences with intracellular $Ca^{2+}$ dynamics, which might be mediated by an indirect effect via the $H_2O_2$-sensentive TRPM2 channels. Dapagliflozin has no effects on ER calcium release, SOCE, ORAI channels, TRPC4, and TRPC5 channels. These findings give a new insight into the pharmacology of SGLT-2 inhibitors.

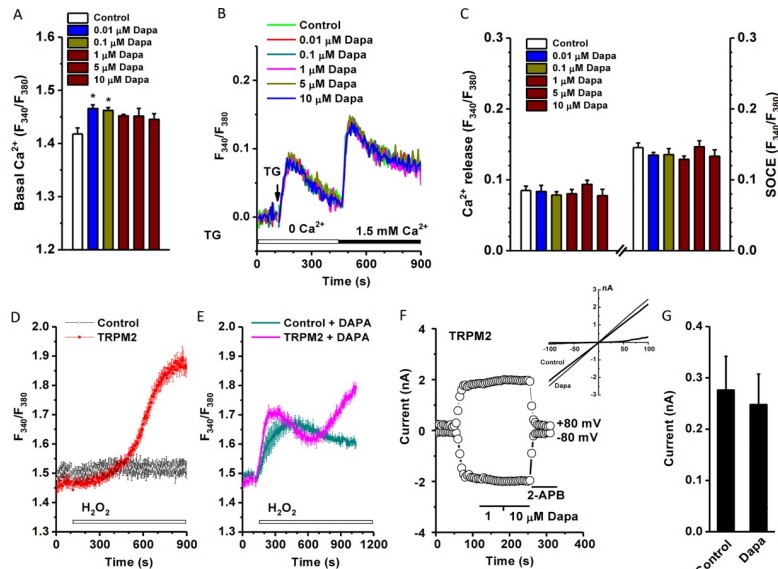

**Fig 7. Effect of dapagliflozin on basal $[Ca^{2+}]_i$ and $Ca^{2+}$ influx in HK-2-cells.** HK-2 cells were seeded at a confluence of $6 \times 10^4$ cells/well and incubated for 12 hours before loading with Fura-PE3/AM (1 μM). The ratio ($F_{340}/F_{380}$) was measured using FlexStation 3. A, Mean ± SEM data for the basal $Ca^{2+}$ level ($n = 7$ for each group). B, Effect on thapsigargin (TG, 1 μM) induced $Ca^{2+}$ release and store-operated $Ca^{2+}$ entry (SOCE) after refilling with 1.5 mM $Ca^{2+}$. C, Mean ± SEM data for (B) ($n = 7$). D, HEK293 T-REx cells over-expressing with human TRPM2 and the activation by 1 mM $H_2O_2$ ($n = 8$). E, Dapagliflozin (Dapa) altered intracellular $Ca^{2+}$ dynamics related to the TRPM2 channels. F, No direct effect on TRPM2 channel activated by ADP-ribose (0.5 mM). G, Mean ± SEM data for (F) ($n = 4$).

Hydrogen peroxide has been used *in vitro* cell models to mimic oxidative stress conditions. The concentration at 100–500 μM has been used in different cell types to induce apoptotic cell death, necrosis and the inhibition of cell proliferation [15,30]. We have examined the dose response of $H_2O_2$ in the HK-2 cells and found that the concentration near $IC_{50}$ (200 μM) can successfully mimic oxidative stress condition. Dapagliflozin at 0.1 to 5 μM shows a significant protective effect against oxidative stress-induced cell injury. Both $H_2O_2$-induced apoptosis and cell growth inhibition are alleviated by dapagliflozin. However high concentrations of dapagliflozin (more than 10 μM) may cause cytotoxicity. Based on human pharmacokinetics data, the maximum therapeutic plasma concentration of dapagliflozin ($C_{max}$) is 158 ng/ml (~0.39 μM) [3] and the extreme high $C_{max}$ is 1000 ng/ml (~2.44 μM) [31], therefore the protective effect of dapagliflozin at 0.1 to 1 μM should be considered as a therapeutic effect.

The real-time monitoring of fluorescence intensity of specific ROS indicators allows us to accurately detect cytosolic ROS or mitochondria ROS production. The control group and test groups were run in parallel in same plate, which produces more reliable data for comparison. We found that the cytosolic ROS is enhanced under oxidative stress and dapagliflozin at 0.1 μM reduced the enhanced ROS production. This finding is consistent with the *in vivo* study in Akita mice kidney and the *in vitro* study using cultured murine proximal tubular epithelial cells under high glucose condition [10,32]. We have also examined the mitochondrial ROS production using MitoSOX™ red. MitoSOX is cell permeable and rapidly oxidized by superoxide but not by other ROS and reactive nitrogen species (RNS). In order to confirm the dye specificity, the mitochondrial chain uncoupler FCCP was used as a positive control in this study, which has been reported to increase MitoSOX fluorescence intensity [33]. Our data show that the prevention of ROS production by dapagliflozin (0.1 μM) is seen in the oxidative stress condition, but not in the normal condition. Too high concentrations (≥ 10 μM) of

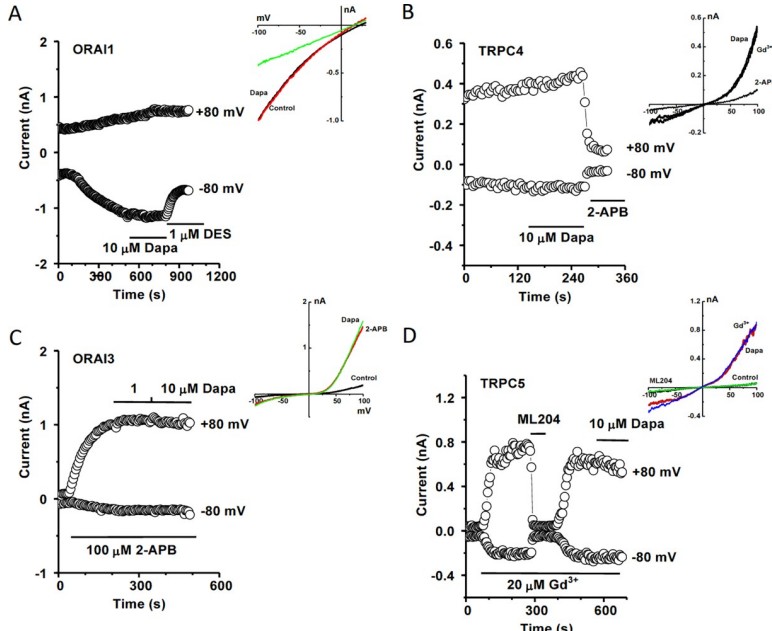

**Fig 8. No effect of dapagliflozin on ORAI1, ORAI3, TRPC4 and TRPC5 currents.** Whole-cell patch recording was performed using HEK293 T-REx cells overexpressing ORAI or TRPC channels. The detail procedures are similar to our previous reports [13,27–29]. A, Example of ORAI1 current recorded in the HEK293 T-REx cells expressing ORAI1-mCherry. The current was induced by thapsigargin (1 μM) in the pipette solution. Dapagliflozin (Dapa) showed no obvious effect on the current, but diethylstilbestrol (DES) as a positive control blocker significantly inhibited the ORAI1 current. B, Example of ORAI3 current induced by 2-APB (100 μM) and the effect of Dapa (1, 10 μM). C, Example of TRPC4 current in the presence of activator $Gd^{3+}$ (20 μM) and the effect of Dapa (10 μM). 2-APB (100 μM) was added as a positive control (inhibitor). D, TRPC5 was induced by $Gd^{3+}$ (20 μM) and the effect of ML204 (10 μM) and Dapa (10 μM). The IV curves were shown as insets in each panel. N = 3–5 for each channel current recorded by whole-cell patch.

dapagliflozin have no protective effects rather than cytotoxicity. It is unclear how dapagliflozin alters the mitochondrial activity. Dapagliflozin may influence on Bcl2 or Bax expression [34]. Dapagliflozin may also cause a modest inhibition of mitochondrial complex I and activate AMPK [35]. The anti-oxidant capacity of gliflozin family has been examined using an oxygen radical absorbance capacity assay, dapagliflozin itself has less anti-oxidant capacity by its chemical structure comparing with remogliflozin and canagliflozin [36], but strong antioxidant effect was seen in cells, suggesting the effect is due to a biological process.

Oxidative stress and high glucose in diabetes are closely related to intracellular $Ca^{2+}$ dynamics, calcium channels and apoptotic protein regulators [13,37–39]. Therefore, we have examined the $Ca^{2+}$ release, $Ca^{2+}$ influx and some $Ca^{2+}$ channels. Dapagliflozin slightly increases the basal $Ca^{2+}$ level in HK-2 cells, but does not change $Ca^{2+}$ release and store-operated $Ca^{2+}$ influx. However, the $Ca^{2+}$ influx via TRPM2 seems to be facilitated by dapagliflozin when $H_2O_2$ is applied, which could be due to the dynamic change of intracellular $Ca^{2+}$ homeostasis and then affect TRPM2 channel sensitivity to $[Ca^{2+}]_i$ [38]. Nevertheless, the channel itself did not show any direct response to dapagliflozin in the patch clamp experiments, where the intracellular $Ca^{2+}$ level was clamped with the buffered pipette solution. Dapagliflozin may have a differential effect from other SGLT-2 inhibitors, since empagliflozin and canagliflozin can cause coronary vasodilation, but dapagliflozin cannot [40]. In addition, the inhibition of $Na^+$-$H^+$ exchanger (NHX) by SGLT-2 inhibitors may also affect the mitochondrial and cytosol $Ca^{2+}$ homeostasis [12,41,42]. SGLT-2 is almost exclusively expressed in the kidney proximal tubules

although it has also been reported in other organs, such as pancreatic alpha cells [43], therefore it could be difficult to extend the concept from the cell types with low SGLT-2 and high L-type $Ca^{2+}$ channels expression, such as cardiac myocytes, to the HK-2 cells with high SGLT-2 expression and no L-type $Ca^{2+}$ channels, since dapagliflozin inhibits L-type channel expression in the heart [44].

In conclusion, dapagliflozin displays a significant protective effect against oxidative stress induced proximal tubular cell injury. This protective effect is mediated by the reduction of cytosolic ROS and mitochondrial superoxide ROS production. Dapagliflozin also alters the dynamics of $Ca^{2+}$ influx via the oxidative-sensitive TRPM2 $Ca^{2+}$ channels. The antioxidative effect could be a new alternative explanation for the beneficial effects of dapagliflozin.

## Supporting information

**S1 Fig. The full PCR gel and the cropped area presented in Fig 1.**
(PPTX)

## Acknowledgments

This work was supported by the European Union's Horizon 2020 research and innovation programme and EFPIA with JDRF (to S.Z.X.). P.L. received the scholarship from China Scholarship Council as a visiting scholar to Hull York Medical School.

## Author Contributions

**Conceptualization:** Nawel Zaibi, Pengyun Li, Shang-Zhong Xu.

**Data curation:** Nawel Zaibi.

**Formal analysis:** Nawel Zaibi, Shang-Zhong Xu.

**Funding acquisition:** Shang-Zhong Xu.

**Investigation:** Nawel Zaibi, Pengyun Li.

**Methodology:** Nawel Zaibi, Pengyun Li.

**Project administration:** Shang-Zhong Xu.

**Supervision:** Shang-Zhong Xu.

**Validation:** Nawel Zaibi, Pengyun Li.

**Writing – original draft:** Shang-Zhong Xu.

**Writing – review & editing:** Shang-Zhong Xu.

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
