## [Decision Letter · Decision Letter 0]

29 Sep 2020

PONE-D-20-22450

Protective effects of dapagliflozin against oxidative stress-induced cell injury in human proximal tubular cells

PLOS ONE

Dear Dr. Xu,

Thank you for submitting your manuscript to PLOS ONE. After careful consideration, we feel that it has merit but does not fully meet PLOS ONE’s publication criteria as it currently stands. Therefore, we invite you to submit a revised version of the manuscript that addresses the points raised during the review process.

For this manuscript to be revised major concerns need to be addressed which include improved presentation/writing of the document, studying the relationship between SGLT proteins and oxidative stress and other concerns raised by the reviewer

We look forward to receiving your revised manuscript.

Kind regards,

Rodrigo Franco

Academic Editor

PLOS ONE

Journal Requirements:

2. Please provide additional information about each of the cell lines used in this work, including the source of the inducible TRPM2 cells and any quality control testing procedures conducted (authentication, characterisation, and mycoplasma testing). For more information, please see http://journals.plos.org/plosone/s/submission-guidelines#loc-cell-lines.

3. At this time, we ask that you please provide scale bars on the microscopy images presented in Figure 2E and 3C and refer to the scale bar in the corresponding Figure legend.

4.We note that you have indicated that data from this study are available upon request. PLOS only allows data to be available upon request if there are legal or ethical restrictions on sharing data publicly. For more information on unacceptable data access restrictions, please see http://journals.plos.org/plosone/s/data-availability#loc-unacceptable-data-access-restrictions.

5.PLOS ONE now requires that authors provide the original uncropped and unadjusted images underlying all blot or gel results reported in a submission’s figures or Supporting Information files. This policy and the journal’s other requirements for blot/gel reporting and figure preparation are described in detail at https://journals.plos.org/plosone/s/figures#loc-blot-and-gel-reporting-requirements and https://journals.plos.org/plosone/s/figures#loc-preparing-figures-from-image-files. When you submit your revised manuscript, please ensure that your figures adhere fully to these guidelines and provide the original underlying images for all blot or gel data reported in your submission. See the following link for instructions on providing the original image data: https://journals.plos.org/plosone/s/figures#loc-original-images-for-blots-and-gels.

6. Please ensure that you refer to Figure 8 in your text as, if accepted, production will need this reference to link the reader to the figure.

7.We suggest you thoroughly copyedit your manuscript for language usage, spelling, and grammar. If you do not know anyone who can help you do this, you may wish to consider employing a professional scientific editing service.  

Reviewers' comments:

Reviewer's Responses to Questions

**Comments to the Author**

1. Is the manuscript technically sound, and do the data support the conclusions?

Reviewer #1: Yes

2. Has the statistical analysis been performed appropriately and rigorously? 

Reviewer #1: No

3. Have the authors made all data underlying the findings in their manuscript fully available?

Reviewer #1: No

4. Is the manuscript presented in an intelligible fashion and written in standard English?

Reviewer #1: No

5. Review Comments to the Author

Reviewer #1: Question 1: In the section of the results, the subtitles should highlight the central idea of the paragraph. Therefore, the results section should be elaborated in more detail.

Question 2: How did the authors choose to study on SGLT1 and SGLT2?

Question 3: The relationship between SGLT1 and SGLT2 and oxidative stress should be clarified.

Question 4: Method: how were the doses of dapagliflozin determined in in vivo? 

Question 5: The author should add in vivo experiments to further verify the role of dapagliflozin in oxidative stress-induced cell injury in human proximal tubular cells.

Question 6: It is unacceptable that the author made a great deal of discussion that they did not present in the results. The authors have to highlight the novelties of their work.

6. PLOS authors have the option to publish the peer review history of their article (what does this mean?). If published, this will include your full peer review and any attached files.

Reviewer #1: No

---

## [Decision Letter · Decision Letter 1]

4 Feb 2021

Protective effects of dapagliflozin against oxidative stress-induced cell injury in human proximal tubular cells

PONE-D-20-22450R1

Dear Dr. Xu,

We’re pleased to inform you that your manuscript has been judged scientifically suitable for publication and will be formally accepted for publication once it meets all outstanding technical requirements.

Kind regards,

Rodrigo Franco

Academic Editor

PLOS ONE

Additional Editor Comments (optional):

Reviewers' comments:

Reviewer's Responses to Questions

**Comments to the Author**

1. If the authors have adequately addressed your comments raised in a previous round of review and you feel that this manuscript is now acceptable for publication, you may indicate that here to bypass the “Comments to the Author” section, enter your conflict of interest statement in the “Confidential to Editor” section, and submit your "Accept" recommendation.

Reviewer #1: All comments have been addressed

2. Is the manuscript technically sound, and do the data support the conclusions?

Reviewer #1: Partly

3. Has the statistical analysis been performed appropriately and rigorously? 

Reviewer #1: Yes

4. Have the authors made all data underlying the findings in their manuscript fully available?

Reviewer #1: Yes

5. Is the manuscript presented in an intelligible fashion and written in standard English?

Reviewer #1: Yes

6. Review Comments to the Author

Reviewer #1: (No Response)

7. PLOS authors have the option to publish the peer review history of their article (what does this mean?). If published, this will include your full peer review and any attached files.

Reviewer #1: No

---

## [Editor Report · Acceptance letter]

9 Feb 2021

PONE-D-20-22450R1 

Protective effects of dapagliflozin against oxidative stress-induced cell injury in human proximal tubular cells  

Dear Dr. Xu:

I'm pleased to inform you that your manuscript has been deemed suitable for publication in PLOS ONE. Congratulations! Your manuscript is now with our production department. 

Kind regards, 

on behalf of

Dr. Rodrigo Franco 

Academic Editor

PLOS ONE